# Real Time Image Saliency for Black Box Classifiers

**Piotr Dabkowski**
pd437@cam.ac.uk
University of Cambridge

**Yarin Gal**
yarin.gal@eng.cam.ac.uk
University of Cambridge
and Alan Turing Institute, London

## Abstract

In this work we develop a fast saliency detection method that can be applied to any differentiable image classifier. We train a masking model to manipulate the scores of the classifier by masking salient parts of the input image. Our model generalises well to unseen images and requires a single forward pass to perform saliency detection, therefore suitable for use in real-time systems. We test our approach on CIFAR-10 and ImageNet datasets and show that the produced saliency maps are easily interpretable, sharp, and free of artifacts. We suggest a new metric for saliency and test our method on the ImageNet object localisation task. We achieve results outperforming other weakly supervised methods.

## 1 Introduction

Current state of the art image classifiers rival human performance on image classification tasks, but often exhibit unexpected and unintuitive behaviour [6, 13]. For example, we can apply a small perturbation to the input image, unnoticeable to the human eye, to fool a classifier completely [13].

Another example of an unexpected behaviour is when a classifier fails to *understand* a given class despite having high accuracy. For example, if "polar bear" is the only class in the dataset that contains snow, a classifier may be able to get a 100% accuracy on this class by simply detecting the presence of snow and ignoring the bear completely [6]. Therefore, even with perfect accuracy, we cannot be sure whether our model actually detects polar bears or just snow. One way to decouple the two would be to find snow-only or polar-bear-only images and evaluate the model's performance on these images separately. An alternative is to use an image of a polar bear with snow from the dataset and apply a *saliency detection method* to test what the classifier is really looking at [6, 11].

Saliency detection methods show which parts of a given image are the most relevant to the model for a particular input class. Such saliency maps can be obtained for example by finding the smallest region whose removal causes the classification score to drop significantly. This is because we expect the removal of a patch which is not useful for the model not to affect the classification score much. Finding such a salient region can be done iteratively, but this usually requires hundreds of iterations and is therefore a time-consuming process.

In this paper we lay the groundwork for a new class of fast and accurate model-based saliency detectors, giving high pixel accuracy and sharp saliency maps (an example is given in figure 1). We propose a fast, model agnostic, saliency detection method. Instead of iteratively obtaining saliency maps for each input image separately, we train a model to predict such a map for any input image in a single feed-forward pass. We show that this approach is not only orders-of-magnitude faster than iterative methods, but it also produces higher quality saliency masks and achieves better localisation results. We assess this with standard saliency benchmarks and introduce a new saliency measure. Our proposed model is able to produce real-time saliency maps, enabling new applications such as video-saliency which we comment on in our *Future Research* section.

## 2 Related work

Since the rise of CNNs in 2012 [5] numerous methods of image saliency detection have been proposed. One of the earliest such methods is a gradient-based approach introduced in [11] which computes the gradient of the class with respect to the image and assumes that salient regions are at locations

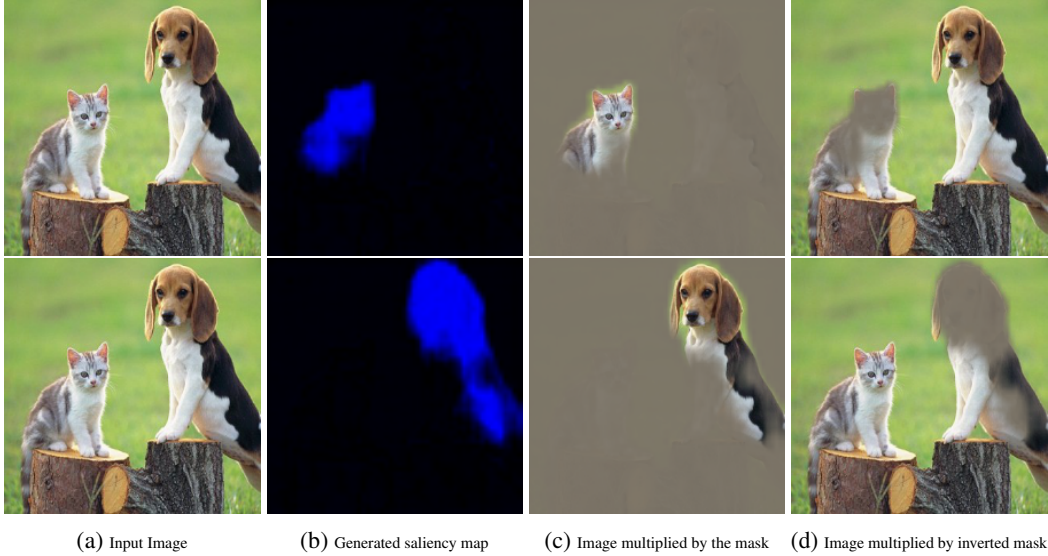

(a) Input Image      (b) Generated saliency map      (c) Image multiplied by the mask      (d) Image multiplied by inverted mask

Figure 1: An example of explanations produced by our model. The top row shows the explanation for the "Egyptian cat" while the bottom row shows the explanation for the "Beagle". Note that produced explanations can precisely both highlight and remove the selected object from the image.

with high gradient magnitude. Other similar backpropagation-based approaches have been proposed, for example Guided Backpropagation [12] or Excitation Backprop [16]. While the gradient-based methods are fast enough to be applied in real-time, they produce explanations of limited quality [16] and they are hard to improve and build upon.

Zhou et al. [17] proposed an approach that iteratively removes patches of the input image (by setting them to the mean colour) such that the class score is preserved. After a sufficient number of iterations, we are left with salient parts of the original image. The maps produced by this method are easily interpretable, but unfortunately, the iterative process is very time consuming and not acceptable for real-time saliency detection.

In another work, Cao et al. [1] introduced an optimisation method that aims to preserve only a fraction of network activations such that the class score is maximised. Again, after the iterative optimisation process, only activations that are relevant remain and their spatial location in the CNN feature map indicate salient image regions.

Very recently (and in parallel to this work), another optimisation based method was proposed [2]. Similarly to Cao et al. [1], Fong and Vedaldi [2] also propose to use gradient descent to optimise for the salient region, but the optimisation is done only in the image space and the classifier model is treated as a black box. Essentially Fong and Vedaldi [2]'s method tries to remove as little from the image as possible, and at the same time to reduce the class score as much as possible. A removed region is then a minimally salient part of the image. This approach is model agnostic and the produced maps are easily interpretable because the optimisation is done in the image space and the model is treated as a black box.

We next argue what conditions a good saliency model should satisfy, and propose a new metric for saliency.

## 3    Image Saliency and Introduced Evidence

Image saliency is relatively hard to define and there is no single obvious metric that could measure the quality of the produced map. In simple terms, the saliency map is defined as a summarised explanation of where the classifier "looks" to make its prediction.

There are two slightly more formal definitions of saliency that we can use:

- Smallest sufficient region (SSR) — smallest region of the image that alone allows a confident classification,

- Smallest destroying region (SDR) — smallest region of the image that when removed, prevents a confident classification.

Similar concepts were suggested in [2]. An example of SSR and SDR is shown in figure 2. It can be seen that SSR is very small and has only one seal visible. Given this SSR, even a human would find it difficult to recognise the preserved image. Nevertheless, it contains some characteristic for "seal" features such as parts of the face with whiskers, and the classifier is over 90% confident that this image should be labeled as a "seal". On the other hand, SDR has a much stronger and larger region and quite successfully removes all the evidence for seals from the image. In order to be as informative as possible, we would like to find a region that performs well as both SSR and SDR.

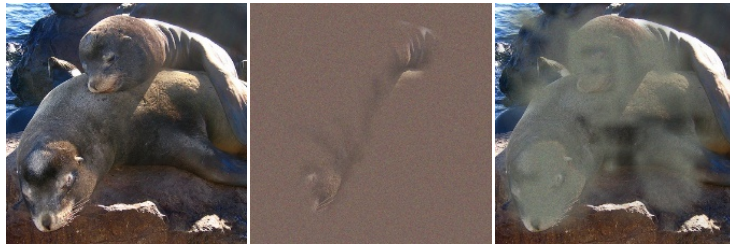

Figure 2: From left to right: the input image; smallest sufficient region (SSR); smallest destroying region (SDR). Regions were found using the mask optimisation procedure from [2].

Both SDR and SSR remove some evidence from the image. There are few ways of removing evidence, for example by blurring the evidence, setting it to a constant colour, adding noise, or by completely cropping out the unwanted parts. Unfortunately, each one of these methods introduces new evidence that can be used by the classifier as a side effect. For example, if we remove a part of the image by setting it to the constant colour green then we may also unintentionally provide evidence for "grass" which in turn may increase the probability of classes appearing often with grass (such as "giraffe"). We discuss this problem and ways of minimising introduced evidence next.

## 3.1 Fighting the Introduced Evidence

As mentioned in the previous section, by manipulating the image we always introduce some extra evidence. Here, let us focus on the case of applying a mask $M$ to the image $X$ to obtain the edited image $E$. In the simplest case we can simply multiply $X$ and $M$ element-wise:

$$E = X \odot M \tag{1}$$

This operation sets certain regions of the image to a constant "0" colour. While setting a larger patch of the image to "0" may sound rather harmless (perhaps following the assumption that the mean of all colors carries very little evidence), we may encounter problems when the mask $M$ is not *smooth*. The mask $M$, in the worst case, can be used to introduce a large amount of additional evidence by generating adversarial artifacts (a similar observation was made in [2]). An example of such a mask is presented in figure 3. Adversarial artifacts generated by the mask are very small in magnitude and almost imperceivable for humans, but they are able to completely destroy the original prediction of the classifier. Such adversarial masks provide very poor saliency explanations and therefore should be avoided.

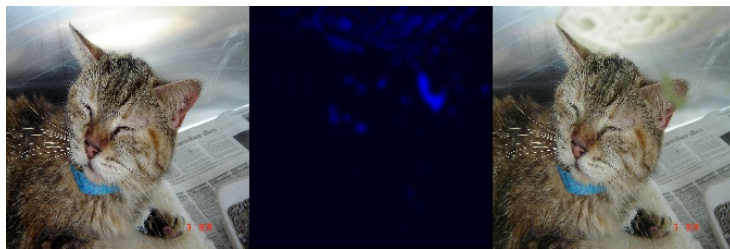

Figure 3: The adversarial mask introduces very small perturbations, but can completely alter the classifier's predictions. From left to right: an image which is correctly recognised by the classifier with a high confidence as a "tabby cat"; a generated adversarial mask; an original image after application of the mask that is no longer recognised as a "tabby cat".

There are a few ways to make the introduction of artifacts harder. For example, we may change the way we apply a mask to reduce the amount of unwanted evidence due to specifically-crafted masks:

$$E = X \odot M + A \odot (1 - M) \tag{2}$$

where $A$ is an alternative image. $A$ can be chosen to be for example a highly blurred version of $X$. In such case mask $M$ simply selectively adds blur to the image $X$ and therefore it is much harder to generate high-frequency-high-evidence artifacts. Unfortunately, applying blur does not eliminate existing evidence very well, especially in the case of images with low spatial frequencies like a seashore or mountains.

Another reasonable choice of $A$ is a random constant colour combined with high-frequency noise. This makes the resulting image $E$ more unpredictable at regions where $M$ is low and therefore it is slightly harder to produce a reliable artifact.

Even with all these measures, adversarial artifacts may still occur and therefore it is necessary to encourage smoothness of the mask $M$ for example via a total variation (TV) penalty. We can also directly resize smaller masks to the required size as resizing can be seen as a smoothness mechanism.

## 3.2 A New Saliency Metric

A standard metric to evaluate the quality of saliency maps is the localisation accuracy of the saliency map. However, it should be noted that saliency is *not* equivalent to localisation. For example, in order to recognise a dog we usually just need to see its head; legs and body are mostly irrelevant for the recognition process. Therefore, saliency map for a dog will usually only include its head while the localisation box always includes a whole dog with not-salient details like legs and tail. The saliency of the object highly overlaps with its localisation and therefore localisation accuracy still serves as a useful metric, but in order to better assess the quality and interpretability of the produced saliency maps, we introduce a new, highly tuned metric.

According to the SSR objective, we require that the classifier is able to still recognise the object from the produced saliency map and that the preserved region is as small as possible. In order to make sure that the preserved region is free from adversarial artifacts, instead of masking we can crop the image. We propose to find the tightest rectangular crop that *contains the entire salient region* and to feed that rectangular region to the classifier to directly verify whether it is able to recognise the requested class. We define our saliency metric simply as:

$$s(a, p) = \log(\tilde{a}) - \log(p) \tag{3}$$

with $\tilde{a} = \max(a, 0.05)$. Here $a$ is the area of the rectangular crop as a fraction of the total image size and $p$ is the probability of the requested class returned by the classifier based on the cropped region. The metric is almost a direct translation of the SSR. We threshold the area at $0.05$ in order to prevent instabilities at low area fractions. Good saliency detectors will be able to significantly reduce the crop size without reducing the classification probability, and therefore a low value for the saliency metric is a characteristic of good saliency detectors.

Interpreting this metric following *information theory*, this measure can be seen as the relative amount of information between an indicator variable with probability $p$ and an indicator variable with probability $a$ — or the *concentration of information* in the cropped region.

Because most image classifiers accept only images of a fixed size and the crop can have an arbitrary size, we resize the crop to the required size disregarding aspect ratio. This seems to work well in practice, but it should be noted that the proposed saliency metric works best with classifiers that are largely invariant to the scale and aspect ratio of the object.

## 3.3 The Saliency Objective

Taking the previous conditions into consideration, we want to find a mask $M$ that is smooth and performs well at both SSR and SDR; examples of such masks can be seen in figure 1. Therefore, more formally, given class $c$ of interest, and an input image $X$, to find a saliency map $M$ for class $c$, our objective function $L$ is given by:

$$L(M) = \lambda_1 TV(M) + \lambda_2 AV(M) - \log(f_c(\Phi(X, M))) + \lambda_3 f_c(\Phi(X, 1 - M))^{\lambda_4} \tag{4}$$

where $f_c$ is a softmax probability of the class $c$ of the black box image classifier and $TV(M)$ is the total variation of the mask defined simply as:

$$TV(M) = \sum_{i,j}(M_{ij} - M_{ij+1})^2 + \sum_{i,j}(M_{ij} - M_{i+1j})^2, \tag{5}$$

$AV(M)$ is the average of the mask elements, taking value between 0 and 1, and $\lambda_i$ are regularisers. Finally, the function $\Phi$ removes the evidence from the image as introduced in the previous section:

$$\Phi(X, M) = X \odot M + A \odot (1 - M). \qquad (6)$$

In total, the objective function is composed of 4 terms. The first term enforces mask smoothness, the second term encourages that the region is small. The third term makes sure that the classifier is able to recognise the selected class from the preserved region. Finally, the last term ensures that the probability of the selected class, after the salient region is removed, is low (note that the inverted mask $1 - M$ is applied). Setting $\lambda_4$ to a value smaller than 1 (e.g. 0.2) helps reduce this probability to very small values.

## 4 Masking Model

The mask can be found iteratively for a given image-class pair by directly optimising the objective function from equation 4. In fact, this is the method used by [2] which was developed in parallel to this work, with the only difference that [2] only optimises the mask iteratively and for SDR (so they don't include the third term of our objective function). Unfortunately, iteratively finding the mask is not only very slow, as normally more than 100 iterations are required, but it also causes the mask to greatly overfit to the image and a large TV penalty is needed to prevent adversarial artifacts from forming. Therefore, the produced masks are blurry, imprecise, and overfit to the specific image rather than capturing the general behaviour of the classifier (see figure 2).

For the above reasons, we develop a trainable masking model that can produce the desired masks in a single forward pass without direct access to the image classifier after training. The masking model receives an image and a class selector as inputs and learns to produce masks that minimise our objective function (equation 4). In order to succeed at this task, the model must learn which parts of the input image are considered salient by the black box classifier. In theory, the model can still learn to develop adversarial masks that perform well on the objective function, but in practice it is not an easy task, because the model itself acts as some sort of a "regulariser" determining which patterns are more likely and which are less.

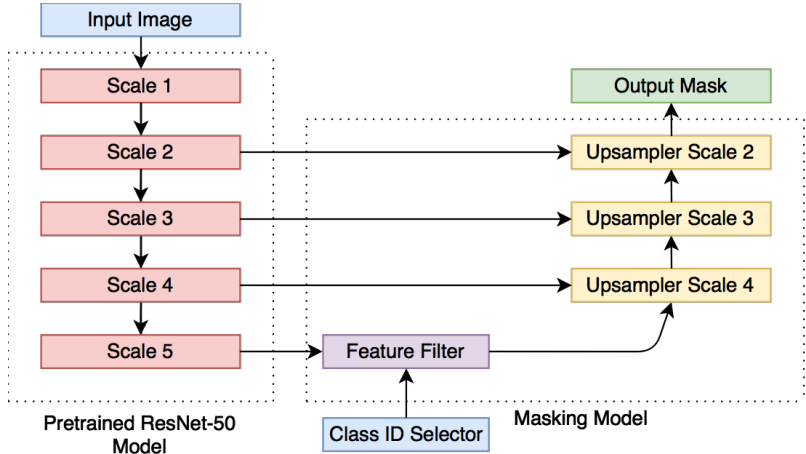

Figure 4: Architecture diagram of the masking model.

In order to make our masks sharp and precise, we adopt a U-Net architecture [8] so that the masking model can use feature maps from multiple resolutions. The architecture diagram can be seen in figure 4. For the encoder part of the U-Net we use ResNet-50 [3] pre-trained on ImageNet [9]. It should be noted that our U-Net is just a model that is trained to predict the saliency map for the given black-box classifier. We use a pre-trained ResNet as a part of this model in order to speed up the training, however, as we show in our CIFAR-10 experiment in section 5.3 the masking model can also be trained completely from scratch.

The ResNet-50 model contains feature maps of five different scales, where each subsequent scale block downsamples the input by a factor of two. We use the ResNet's feature map from Scale 5 (which corresponds to downsampling by a factor of 32) and pass it through the feature filter. The purpose of the feature filter is to attenuate spatial locations which contents do not correspond to

the selected class. Therefore, the feature filter performs the initial localisation, while the following upsampling blocks fine-tune the produced masks. The output of the feature filter $Y$ at spatial location $i, j$ is given by:

$$Y_{ij} = X_{ij}\sigma(X_{ij}^T C_s) \tag{7}$$

where $X_{ij}$ is the output of the Scale 5 block at spatial location $i, j$; $C_s$ is the embedding of the selected class $s$ and $\sigma(\cdot)$ is the sigmoid nonlinearity. Class embedding $C$ can be learned as part of the overall objective.

The upsampler blocks take the lower resolution feature map as input and upsample it by a factor of two using transposed convolution [15], afterwards they concatenate the upsampled map with the corresponding feature map from ResNet and follow that with three bottleneck blocks [3].

Finally, to the output of the last upsampler block (Upsampler Scale 2) we apply 1x1 convolution to produce a feature map with just two channels — $C_0, C_1$. The mask $M_s$ is obtained from:

$$M_s = \frac{\text{abs}(C_0)}{\text{abs}(C_0) + \text{abs}(C_1)} \tag{8}$$

We use this nonstandard nonlinearity because sigmoid and tanh nonlinearities did not optimise properly and the extra degree of freedom from two channels greatly improved training. The mask $M_s$ has resolution four times lower than the input image and has to be upsampled by a factor of four with bilinear resize to obtain the final mask $M$.

The complexity of the model is comparable to that of ResNet-50 and it can process more than a hundred 224x224 images per second on a standard GPU (which is sufficient for real-time saliency detection).

## 4.1 Training process

We train the masking model to directly minimise the objective function from equation 4. The weights of the pre-trained ResNet encoder (red blocks in figure 4) are kept fixed during the training.

In order to make the training process work properly, we introduce few optimisations. First of all, in the naive training process, the ground truth label would always be supplied as a class selector. Unfortunately, under such setting, the model learns to completely ignore the class selector and simply always masks the *dominant object* in the image. The solution to this problem is to sometimes supply a class selector for a fake class and to apply only the area penalty term of the objective function. Under this setting, the model must pay attention to the class selector, as the only way it can reduce loss in case of a fake label is by setting the mask to zero. During training, we set the probability of the fake label occurrence to 30%. One can also greatly speed up the embedding training by ensuring that the maximal value of $\sigma(X_{ij}^T C_s)$ from equation 7 is high in case of a correct label and low in case of a fake label.

Finally, let us consider again the evidence removal function $\Phi(X, M)$. In order to prevent the model from adapting to any single evidence removal scheme the alternative image $A$ is randomly generated every time the function $\Phi$ is called. In 50% of cases the image $A$ is the blurred version of $X$ (we use a Gaussian blur with $\sigma = 10$ to achieve a strong blur) and in the remainder of cases, $A$ is set to a random colour image with the addition of a Gaussian noise. Such a random scheme greatly improves the quality of the produced masks as the model can no longer make strong assumptions about the final look of the image.

## 5 Experiments

In the ImageNet saliency detection experiment we use three different black-box classifiers: AlexNet [5], GoogleNet [14] and ResNet-50 [3]. These models are treated as black boxes and for each one we train a separate masking model. The selected parameters of the objective function are $\lambda_1 = 10$, $\lambda_2 = 10^{-3}$, $\lambda_3 = 5$, $\lambda_4 = 0.3$. The first upsampling block has 768 output channels and with each subsequent upsampling block we reduce the number of channels by a factor of two. We train each masking model as described in section 4.1 on 250,000 images from the ImageNet training set. During the training process, a very meaningful class embedding was learned and we include its visualisation in the Appendix.

Example masks generated by the saliency models trained on three different black box image classifiers can be seen in figure 5, where the model is tasked to produce a saliency map for the ground truth

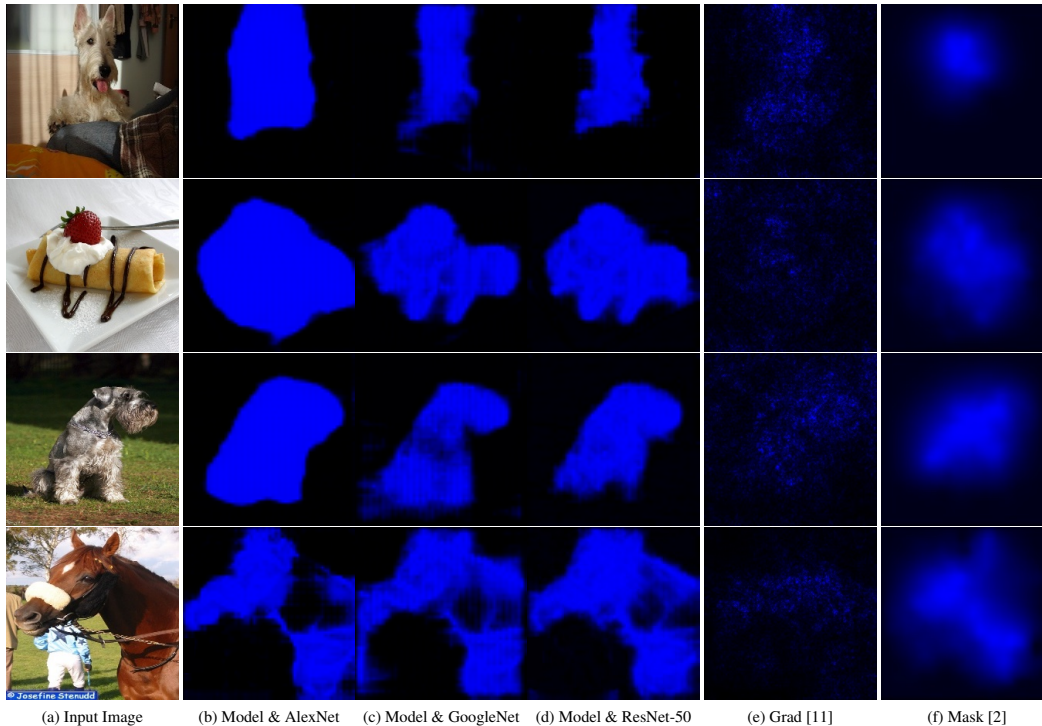

| (a) Input Image | (b) Model & AlexNet | (c) Model & GoogleNet | (d) Model & ResNet-50 | (e) Grad [11] | (f) Mask [2] |

Figure 5: Saliency maps generated by different methods for the ground truth class. The ground truth classes, starting from the first row are: Scottish terrier, chocolate syrup, standard schnauzer and sorrel. Columns b, c, d show the masks generated by *our* masking models, each trained on a different black box classifier (from left to right: AlexNet, GoogleNet, ResNet-50). Last two columns e, f show saliency maps for GoogleNet generated respectively by gradient [11] and the recently introduced iterative mask optimisation approach [2].

label. In figure 5 it can be clearly seen that the quality of masks generated by our models clearly outperforms alternative approaches. The masks produced by models trained on GoogleNet and ResNet are sharp and precise and would produce accurate object segmentations. The saliency model trained on AlexNet produces much stronger and slightly larger saliency regions, possibly because AlexNet is a less powerful model which needs more evidence for successful classification. Additional examples can be seen in the appendix A.

## 5.1 Weakly supervised object localisation

As discussed in section 3.2 a standard method to evaluate produced saliency maps is by object localisation accuracy. It should be noted that our model was not provided any localisation data during training and was trained using only image-class label pairs (weakly supervised training).

We adopt the localisation accuracy evaluation protocol from [1] and provide the ground truth label to the masking model. Afterwards, we threshold the produced saliency map at $0.5$ and the tightest bounding box that contains the whole saliency map is set as the final localisation box. The localisation box has to have IOU greater than $0.5$ with any of the ground truth bounding boxes in order to consider the localisation successful, otherwise, it is counted as an error. The calculated error rates for the three models are presented in table 1. The lowest localisation error of $36.7\%$ was achieved by the saliency model trained on the ResNet-50 black box, this is a good achievement considering the fact that our method was not given any localisation training data and that a fully supervised approach employed by VGG [10] achieved only slightly lower error of $34.3\%$. The localisation error of the model trained on GoogleNet is very similar to the one trained on ResNet. This is not surprising because both models produce very similar saliency masks (see figure 5). The AlexNet trained model, on the other hand, has a considerably higher localisation error which is probably a result of AlexNet needing larger image contexts to make a successful prediction (and therefore producing saliency masks which are slightly less precise).

We also compared our object localisation errors to errors achieved by other weakly supervised methods and existing saliency detection techniques. As a baseline we calculated the localisation error

|                       | Alexnet [5] | GoogleNet [14] | ResNet-50 [3] |
|-----------------------|-------------|----------------|---------------|
| Localisation Err (%)  | 39.8        | 36.9           | **36.7**      |

Table 1: Weakly supervised bounding box localisation error on ImageNet validation set for our masking models trained with different black box classifiers.

of the centrally placed rectangle which spans half of the image area — which we name "Center". The results are presented in table 2. It can be seen that our model outperforms other approaches, sometimes by a significant margin. It also performs significantly better than the baseline (centrally placed box) and iteratively optimised saliency masks. Because a big fraction of ImageNet images have a large, dominant object in the center, the localisation accuracy of the centrally placed box is relatively high and it managed to outperform two methods from the previous literature.

| Center | Grad [11] | Guid [12] | CAM [18] | Exc [16] | Feed [1] | Mask [2] | This Work |
|--------|-----------|-----------|----------|----------|----------|----------|-----------|
| 46.3   | 41.7      | 42.0      | 48.1     | 39.0     | 38.7     | 43.1     | **36.9**  |

Table 2: Localisation errors(%) on ImageNet validation set for popular weakly supervised methods. Error rates were taken from [2] which recalculated originally reported results using few different mask thresholding techniques and achieved slightly lower error rates. For a fair comparison, all the methods follow the same evaluation protocol of [1] and produce saliency maps for GoogleNet classifier [14].

## 5.2 Evaluating the saliency metric

To better assess the interpretability of the produced masks we calculate the saliency metric introduced in section 3.2 for selected saliency methods and present the results in the table 3. We include a few baseline approaches — the "Central box" introduced in the previous section, and the "Max box" which simply corresponds to a box spanning the whole image. We also calculate the saliency metric for the ground truth bounding boxes supplied with the data, and in case the image contains more than one ground truth box the saliency metric is set as the average over all the boxes.

Table 3 shows that our model achieves a considerably better saliency metric than other saliency approaches. It also significantly outperforms max box and center box baselines and is on par with ground truth boxes which supports the claim that the interpretability of the localisation boxes generated by our model is similar to that of the ground truth boxes.

|                              | Localisation Err (%) | Saliency Metric |
|------------------------------|----------------------|-----------------|
| Ground truth boxes (baseline)| 0.00                 | 0.284           |
| Max box (baseline)           | 59.7                 | 1.366           |
| Center box (baseline)        | 46.3                 | 0.645           |
| Grad [11]                    | 41.7                 | 0.451           |
| Exc [16]                     | 39.0                 | 0.415           |
| Masking model (this work)    | **36.9**             | **0.318**       |

Table 3: ImageNet localisation error and the saliency metric for GoogleNet.

## 5.3 Detecting saliency of CIFAR-10

To verify the performance of our method on a completely different dataset we implemented our saliency detection model for the CIFAR-10 dataset [4]. Because the architecture described in section 4 specifically targets high-resolution images and five downsampling blocks would be too much for 32x32 images, we modified the architecture slightly and replaced the ResNet encoder with just 3 downsampling blocks with 5 convolutional layers each. We also reduced the number of bottleneck blocks in each upsampling block from 3 to 1. Unlike before, with this experiment, we did not use a pre-trained masking model, but instead a randomly initialised one. We used a FitNet [7] trained to 92% validation accuracy as a black box classifier to train the masking model. All the training parameters were used following the ImageNet model.

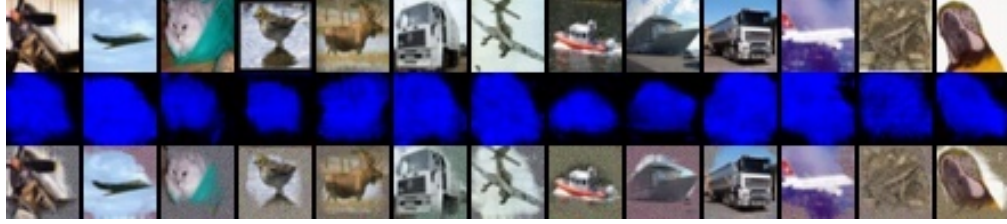

Figure 6: Saliency maps generated by our model for images from CIFAR-10 validation set.

The masking model was trained for 20 epochs. Saliency maps for sample images from the validation set are shown in figure 6. It can be seen that the produced maps are clearly interpretable and a human could easily recognise the original objects after masking. This confirms that the masking model works as expected even at low resolution and that FitNet model, used as a black box learned correct representations for the CIFAR-10 classes. More interestingly, this shows that the masking model does not need to rely on a pre-trained model which might inject its own biases into the generated masks.

# 6   Conclusion and Future Research

In this work, we have presented a new, fast, and accurate saliency detection method that can be applied to any differentiable image classifier. Our model is able to produce 100 saliency masks per second, sufficient for real-time applications. We have shown that our method outperforms other weakly supervised techniques at the ImageNet localisation task. We have also developed a new saliency metric that can be used to assess the quality of explanations produced by saliency detectors. Under this new metric, the quality of explanations produced by our model outperforms other popular saliency detectors and is on par with ground truth bounding boxes.

The model-based nature of our technique means that our work can be extended by improving the architecture of the masking network, or by changing the objective function to achieve any desired properties for the output mask.

Future work includes modifying the approach to produce high quality, weakly supervised, image segmentations. Moreover, because our model can be run in real-time, it can be used for video saliency detection to instantly explain decisions made by black-box classifiers such as the ones used in autonomous vehicles. Lastly, our model might have biases of its own — a fact which does not seem to influence the model performance in finding biases in other black boxes according to the various metrics we used. It would be interesting to study the biases embedded into our masking model itself, and see how these affect the generated saliency masks.

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
