[Supplementary Material]

# A  Appendix

Figure 7 shows a t-SNE visualisation of the embedding learned by the masking model trained on the ImageNet. It can be clearly seen that closely related objects have similar localisations in the embedding. For example fungi and geographical formations, they both form their own clusters. Figure 8 shows a subset of the embedding and again it can be clearly seen that similar dogs occupy similar positions (for example Labrador Retriever and Golden Retriever).

Figure 7: T-SNE visualisation of the class embedding learned by the masking model.

Figure 8: T-SNE visualisation of the class embedding learned by the masking model.

Figure 9: Masks generated by our model for the selected target class. Notice how the cat is masked in the third image because it does not contribute to the selected class - desk.

Figure 10: Masks generated by our model for the selected target class. Note that no mask was generated for the first image because the selected target class (Irish setter) is not present in the image.

Figure 11: Masks generated by our model for the selected target class. Notice that in the first and second image the classifier apparently needs more evidence to be able to recognise classes like ski or bearskin. It makes sense because it would be very hard to recognise these classes if only corresponding objects were masked without supporting evidence.

Figure 12: Masks generated by our model for the selected target class.

Figure 13: Masks generated by our model for the selected target class.

Figure 14: Masks generated by our model for the selected target class. Note that no mask was generated for the third image because the selected target class (street sign) is not present in the image.