[Reviews · NeurIPS 2017]

Reviewer 1



The paper proposes an approach to learn saliency masks. The proposed approach is based on a neural network and can process multiple images per second (i.e. it is fast). To me the paper is borderline, I would not object rejection or acceptance. I really believe in the concept of learning to explain a model and I think the paper has some good ideas. There are no obvious mistakes but there are clear limitations. -The saliency is only indirectly measured. Either through weakly supervised localisation or by the proposed saliency metric both these methods have clear limitations and I think these limitations should be discussed in the paper. The weakly supervised location is not perfect as a measure. If the context in which an object appears is essential to determine its class, the object localisation does not have to correlate with saliency quality. The results on weakly supervised localisation are interesting, but I think there is a big caveat when using them as a quality metric for saliency. The saliency metric is not perfect because of how it is applied. The estimated salient region is cropped. This crop is then rescaled to the original image size, with the original aspect ratio. This could introduce two artefacts. First, the change of the aspect ratio might impact how well it can be classified. Second in the proposed metric a small salient region is preferred. Since a small region is blown up heavily for re-classification the scale at which the object is now presented to the classifier might not be ideal. (Convnets are generally pretty translation invariant, but the scaling invariance must be learned, and there are probably limits to this). What is not discussed here is how much the masking model depends on the architecture for learning the masks? Did the authors at one point experiment with different architectures and how this influenced the result? Minor comments Are the results in Table 1 obtained for all classes or only for the correct class? - Please specify with LRP variant and parameter setting was used for comparison. They have an epsilon, alpha-beta, and more variants with parameters. *** Post rebuttal edit *** The fact that how well the saliency metric works depends on the quality and the scale invariance of the classifier is strongly limiting the applicability of the proposed method. It can only be applied to networks having this invariance. This has important consequences: - The method cannot be used for models during the training phase, nor for models that do not exhibit this invariance. - This limit the applicability to other domains (e.g. spectrogram analysis with CNN's). - The method is not generally applicable to black-box classifiers as claimed in the title. Furthermore, the response hints at a strong dependence on the masking network. - As a result, it is not clear to me whether we are visualizing the saliency of the U-network or the masking network. If these effects are properly discussed in the paper I think it is balanced enough for publication. If not it should not be published

Reviewer 2



This paper introduces a NN to predict the regions of an image that are important for another NN for object categorization. The paper reads well and it is interesting that the experiments show that it works so well compared to previous works. The results are in challenging datasets with state-of-the-art NN. I may suggest to motivate a bit better in which applications one may need real time efficiency. Also, I would have liked to see some insight from an analysis of the learnt salient regions (eg. which object categories exploit biases in the background to recognize the object)

Reviewer 3



This paper image saliency mask generation approach that can process a hundred 224x224 images per second on a standard GPU. Their approach trains a masking model that finds the tightest rectangular crop that contains the entire salient region of a particular requested class by a black box classifier, such as Alexnet, GoogleNet, and ResNet. Their model architecture requires image feature map, such as those by ResNet-50, over different scales. The final scale feature will be passed through a feature filter that performs the initial localisation, while the following upsampling blocks fine-tune the produced masks. Experiment shows that their method outperforms other weakly supervised techniques at the ImageNet localisation task. This paper appears to have sufficient references and related works. Do not completely check. This paper appears to be technically correct. Do not completely check. This paper present a number of intuition and discussion on how they design their approach. This paper's presentation is good. Overall, this paper presents interesting technical results that I am a little concerned about the real time speed claim and applications to real world images. Comments: - Does the processing time for 100 images per second include the image resizing operation? If so, what is the running time for other larger images, such as 640 X 480 images taken from iPhone 6s? - Salient objects in this paper is quite large, what if the requested class object is small in the images? Will 224x224 image be enough? - In Table 3, is there any corresponding metrics for other works in Table 2, such as Feed [2]? MISC - LN 273: "More over, because our model" -> "Moreover, because our model" - LN 151: "difference that [3] only optimise the mask" -> "difference that [3] only optimises the mask"